# Simplifying and Understanding State Space Models with Diagonal Linear RNNs

## Abstract

Sequence models based on linear state spaces (SSMs) have recently emerged as a promising choice of architecture for modeling long range dependencies across various modalities. However, they invariably rely on discretization of a continuous state space, which complicates their presentation and understanding. In this work, we dispose of the discretization step, and propose a model based on vanilla Diagonal Linear RNNs (DLR). We empirically show that, despite being conceptually much simpler, DLR is as performant as previously-proposed SSMs on a variety of tasks and benchmarks including Long Range Arena and raw speech classification. Moreover, we characterize the expressivity of SSMs (including DLR) and attention-based models via a suite of 13 synthetic sequence-to-sequence tasks involving interactions over tens of thousands of tokens, ranging from simple operations, such as shifting an input sequence, to detecting co-dependent visual features over long spatial ranges in flattened images. We find that while SSMs report near-perfect performance on tasks that can be modeled via *few* convolutional kernels, they struggle on tasks requiring *many* such kernels and especially when the desired sequence manipulation is *context-dependent*. Despite these limitations, DLR reaches high performance on two higher-order reasoning tasks LISTOPSSUBTREES and PATHFINDERSEGMENTATION-256 with input lengths $8K$ and $65K$ respectively, and gives encouraging performance on PATHFINDERSEGMENTATION-512 with input length $262K$ for which attention is not a viable choice.

## 1 Introduction

Attention-based models (Vaswani et al., 2017) have been successful across many areas of machine learning (Jumper et al., 2021; Ramesh et al., 2021; Radford et al., 2022). Specifically, Transformers pre-trained on large amounts of unlabelled text via a denoising objective have become the standard in natural language processing, exhibiting impressive amounts of linguistic and world knowledge (Chowdhery et al., 2022). Unfortunately, the $\Omega(L^2)$ complexity of self-attention is prohibitive on tasks where the model is required to capture long-range interactions over various parts of a long input. Recently, Gu et al. (2022a) proposed S4, a model that uses linear state spaces for contextualization instead of attention and delivers remarkable performance on long-range reasoning benchmarks such as Long Range Arena (LRA) (Tay et al., 2021). Subsequently, Gupta et al. (2022) showed that S4 can be simplified by assuming state matrices to be diagonal, while maintaining similar performance, which then led to multiple Diagonal State Space (DSS) models (Mehta et al., 2023; Gu et al., 2022b; Smith et al., 2023). DSS models with interleaved attention layers have also delivered state-of-the-art results on language and code modeling (Mehta et al., 2023) as well as on speech recognition (Saon et al., 2023).

While the aforementioned models are indeed simpler than S4, they are all based on discretizations of continuous state spaces that eventually arrive at a diagonal linear RNN (DLR) whose parameterization differs across the above works. This discretization step complicates presentation and is not immediately accessible to the average ML practitioner unfamiliar with control theory. This naturally raises the question that, if eventually all these models reduce to some parameterization of a DLR, why not use DLRs directly as a starting point? Among the factors that make this challenging is the vanishing gradients problem where the spectral radius of a powered matrix vanishes/explodes with the power (Pascanu et al., 2013) . Past works have attempted

to overcome such issues via normalization (Ba et al., 2016), gating (Wolter & Yao, 2018), and specialized initializations (Voelker et al., 2019). Similarly, for adequate performance, the DSS models initialize their state space parameters via HiPPO theory, which is a mathematical framework for long-range signal propagation (Voelker et al., 2019; Gu et al., 2020).

In this work we propose DLR, a simplification of DSS that directly uses diagonal linear RNNs as a starting point, making it conceptually straightforward to understand and resulting in a cleaner formulation that obviates some unnecessary terms that arise as a by-product of discretization. Unlike traditional RNNs, DLR uses complex-valued transition matrices that, when initialized appropriately with eigenvalues close to the unit circle, can enable signal propagation over a million positions. Moreover, the periodicity of DLR with respect to its parameters suggests a natural initialization at which a DLR of size $N$ can express arbitrary convolutional kernels of length $N$.

We first analyze the expressivity and performance of DLR in a lab setting, along with DSS-based and attention-based models, using a suite of 13 synthetic sequence-to-sequence tasks requiring interactions over tens of thousands of positions and with varying degrees of difficulty ranging from a simple operation, such as shifting a given sequence, to detecting co-dependent visual features over long spatial ranges in flattened images. Synthetic tasks allow us to programmatically generate data, and to experiment with a large range of input lengths. We construct a wide array of atomic tasks for pinpointing skills such as shifting, copying, reversing, and sorting sequences, and for revealing any immediate shortcomings of models and whether, from an expressivity point of view, one model is subsumed by another.

Moreover, two of our proposed tasks are higher-order classification tasks, LISTOPSSUBTREES and PATHFINDERSEGMENTATION, which require multiple skills, such as hierarchical computation and detecting complex long-range spatial relationships in cluttered scenes. Our PATHFINDERSEGMENTATION task is over sequences of lengths $65K$ and $262K$ respectively, which is $4\times$ and $16\times$ longer compared to the challenging PATH-X task from LRA, making our setting considerably harder.

We empirically find that, DLR performs as well as or better than DSS in all our experiments and is as performant as S4/S4D on LRA, illustrating its viability as a long-range model. Second, while SSM layers (DSS,DLR) perform exceptionally well on manipulation tasks such as shifting (or copying data over) extremely long inputs, they struggle on tasks such as reversing, and especially if the desired sequence manipulation is context-dependent. Based on our results, we hypothesize that current SSM layers are potent on tasks requiring arbitrarily complex but only a *few* convolutional kernels, but struggle on tasks requiring *many* kernels. Similarly, they fail on tasks where, even though for each sample just a few kernels should suffice, the kernels that need to be applied to an input vary with the input itself. By their very design, SSM layers such as DLR learn context-independent kernels and hence struggle on such tasks. For example, a DLR layer learns to perfectly shift a $0.5M$ long input by an arbitrary (but input independent) number of positions, but fails when the amount of the desired shift is context dependent. While we find SSMs to be far more compute efficient than attention, on some tasks even deep SSMs do not match the performance of an attention layer, suggesting that deep SSMs do not subsume attention, and that they both offer complimentary benefits.

To better understand how prohibitive the said limitations are on the higher-order tasks, we train multi-layer SSMs and tractable attention baselines on these tasks. Similar to the atomic tasks, DLR outperforms SSM baselines. Surprisingly, while our attention-based baseline is upto $8\times$ slower and far less compute-efficient compared to SSMs, it outperforms them on LISTOPS-SUBTREES . This contradicts the low performance of Transformer variants reported on the LISTOPS task of LRA, highlighting the benefits of our sequence-tagging setup, where supervision is much more dense compared to LRA. PATHFINDERSEGMENTATION, on the other hand, requires contextualization over significantly longer ranges, and here DLR not only outperforms all baselines but delivers an impressive performance on images as large as $256 \times 256$ (input length $65K$) and reasonable performance in the $512 \times 512$ case (input length $262K$).

To summarize, our work comprises several contributions towards better understanding of SSMs. First, we propose DLR, a simpler and equally effective model compared to current SSMs for modeling long-range interactions. Second, we analyze SSMs on a wide range of synthetic tasks and highlight their advantages and limitations compared to other state space and attention-based models. Last, we provide a suite of

synthetic tasks that provide a test-bed for analyzing long-range models. Our code and data are available at `https://anonymized`.

## 2 Method

SSMs such as DSS are based on discretizations of continuous state spaces which makes their presentation complicated and less accessible to the average ML practitioner. We now describe our simplification of DSS that directly uses diagonal linear RNNs as a starting point and does not require any background in control theory. Moreover, removing the discretization step results in a cleaner model, where various scaling factors arising due to discretization are no longer needed. One difference from the traditional RNNs is that we will work over $\mathbb{C}$ instead of $\mathbb{R}$ which, as we will show in §3.3, is important for capturing long-term dependencies.

### 2.1 Diagonal Linear RNN

Parameterized by $\Lambda, w \in \mathbb{C}^N$, a diagonal linear RNN (DLR) defines a 1-dimensional sequence-to-sequence map from an input $(u_0, \ldots, u_{L-1}) = u \in \mathbb{R}^L$ to output $(y_0, \ldots, y_{L-1}) = y \in \mathbb{C}^L$ via the recurrence,

$$x_k = \text{diag}(\Lambda)x_{k-1} + \mathbf{1} \cdot u_k \quad , \quad y_k = \langle w, x_k \rangle \tag{1}$$

where $x_k \in \mathbb{C}^{N \times 1}$, $\text{diag}(\Lambda) \in \mathbb{C}^{N \times N}$ is a diagonal matrix with diagonal $\Lambda$ and $\langle a, b \rangle := \sum_i a_i b_i$. As $\text{diag}(\Lambda)$ is diagonal, the $N$ dimensions of the state $x_k$ do not interact and hence can be computed independently. Assuming $\Lambda = (\lambda_1, \ldots, \lambda_N)$, we obtain the simple recurrence

$$x_{i,k} = \lambda_i x_{i,k-1} + u_k . \tag{2}$$

Assuming $x_{-1} = 0$ for simplicity, Equation 2 can be explicitly unrolled as

$$x_{i,k} = \sum_{j=0}^{k} \lambda_i^j u_{k-j} \quad , \quad y_k = \sum_{j=0}^{k} \langle w, \Lambda^j \rangle u_{k-j} \tag{3}$$

where $\Lambda^j$ is element-wise powered $\Lambda$. For convenience, define the convolutional kernel $K \in \mathbb{C}^L$ as

$$K = (\langle w, \Lambda^k \rangle)_{0 \leqslant k < L} \quad , \quad y_k = \sum_{j=0}^{k} K_j \cdot u_{k-j} . \tag{4}$$

Given an input sequence $u \in \mathbb{R}^L$, one can compute the output $y \in \mathbb{C}^L$ sequentially via the recurrence in Equation 2 but sequential computation on long inputs is prohibitively slow.[1] Instead, Equation 4 can be used to compute all elements of $y$ in parallel after computing $K = w_{1 \times N} \cdot P_{N \times L}$ with $P_{ik} = \lambda_i^k$.

Given an input sequence $u \in \mathbb{R}^L$ and the kernel $K \in \mathbb{C}^L$, naively using Equation 4 for computing $y$ would require $O(L^2)$ multiplications. This can be done much more efficiently in $O(L \log(L))$ time via Fast Fourier Transform (FFT) (see §A.1).

**Casting to $\mathbb{R}$** Equation 1 defines a map $u \in \mathbb{R}^L \mapsto y \in \mathbb{C}^L$ but the produced $y$ needs to be cast to $\mathbb{R}$ for the rest of the layers in the network. We simply cast each $y_k$ to $\mathbb{R}$ as $\text{Re}(y_k)$. Hence, we can assume Equation 4 to be with an explicit cast operation $y_k = \text{Re}(\sum_{j=0}^{k} K_j \cdot u_{k-j})$. As $u$ is over $\mathbb{R}$, this further implies $y_k = \sum_{j=0}^{k} \text{Re}(K_j) \cdot u_{k-j}$. Hence, we can also cast $K \in \mathbb{C}^L$ produced in Equation 4 to $\mathbb{R}$ by taking its real part before computing $y$.

---

[1] Discounted cumulative sum has parallel implementations (Blelloch, 1990) and is supported by some libraries such as JAX as leveraged in the S5 model (Smith et al., 2023). After dropping subscript $i$, Equation 2 can be unrolled as $x_k = \sum_{j=0}^{k} \lambda^{k-j} u_j = \lambda^k \sum_{j=0}^{k} u_j \lambda^{-j} = \lambda^k \sum_{j=0}^{k} \tilde{u}_j$ where $\sum_{j=0}^{k} \tilde{u}_j$ is a vanilla cumulative sum with more extensively supported parallel implementations. Unfortunately, this reduction is numerically stable only if $|\lambda| = 1$ and we instead resort to FFT-based convolution (§A.1) in this work.

In §3 we also experiment with an alternate choice of casting denoted by *DLR-prod* in which instead of casting the kernel elements as $\mathrm{Re}(K_k)$ we use $\mathrm{Re}(K_k) \cdot \mathrm{Im}(K_k)$. In §A.2 we prove that this kernel corresponds to the kernel of a DLR of size at most $4N^2$ and generalize this to define the *Kronecker product* of DLRs where the elementwise product of DLR kernels is shown to correspond to a DLR itself. We will see that this alternative gives significantly better results on tasks such as shifting the elements of a sequence, which can be described in terms of sparse kernels.

**Bidirectional DLR**   In Equation 4, $y_i$ does not depend on $y_{>i}$ and hence the model is left-to-right only. To benefit from bidirectionality, we form a bidirectional version by simply summing the outputs of two independent DLR's, one for each direction:

$$
\begin{aligned}
\overrightarrow{x}_k &= \mathrm{diag}(\Lambda_1)\overrightarrow{x}_{k-1} + \mathbf{1} \cdot u_k \\
\overleftarrow{x}_k &= \mathrm{diag}(\Lambda_2)\overleftarrow{x}_{k-1} + \mathbf{1} \cdot u_{L-1-k} \\
y_k &= \langle w_1, \overrightarrow{x}_k \rangle \; + \; \langle w_2, \overleftarrow{x}_{L-1-(k+1)} \rangle \,.
\end{aligned}
\tag{5}
$$

Similar to Equation 4, we have

$$
\overrightarrow{K} \;=\; (\langle w_1, \Lambda_1^k \rangle)_{0 \leqslant k < L} \;\;,\;\; \overleftarrow{K} \;=\; (\langle w_2, \Lambda_2^k \rangle)_{0 \leqslant k < L}
$$

$$
y_k = \sum_{j=0}^{k} \overrightarrow{K}_{k-j} \cdot u_j + \sum_{j=0}^{L-1-(k+1)} \overleftarrow{K}_{L-1-(k+1)-j} \cdot u_{L-1-j}
\tag{6}
$$

$$
= \sum_{j=0}^{k} \overrightarrow{K}_{k-j} \cdot u_j + \sum_{j=k+1}^{L-1} \overleftarrow{K}_{j-(k+1)} \cdot u_j
$$

which is a standard Toeplitz matrix-vector product and can be computed via FFT by reducing it to a circulant matrix-vector product of size $2L$ as described in §A.1.

**Comparison with DSS**   As stated earlier, SSMs like S4 discretize continuous state spaces to arrive at a DLR. For instance, the $\mathrm{DSS}_{\mathrm{EXP}}$ model (Gupta et al., 2022) discretizes the following state space assuming zero-order hold over intervals of size $\Delta \in \mathbb{R}_{>0}$

$$
\frac{dx}{dt}(t) = \mathrm{diag}(\Lambda)x(t) + \mathbf{1}u(t) \;\;,\;\; y(t) = w \cdot x(t),
$$

which results in the DLR

$$
\begin{aligned}
x_k &= \mathrm{diag}(\exp(\Delta\Lambda))x_{k-1} + \mathbf{1} \cdot u_k \\
y_k &= \langle w((\exp(\Delta\Lambda) - \mathbf{1})/\Lambda, x_k \rangle.
\end{aligned}
$$

Equation 1 looks similar, but removes the parameter $\Delta$ and simplifies the computation of $y_k$ by omitting an additional scaling factor.

## 2.2   DLRs are as expressive as general linear RNNs

In this work, we use *diagonal* linear RNNs for contextualization and it is natural to ask if using a *general* linear RNN instead leads to a more expressive model. In particular, for parameters $A \in \mathbb{C}^{N \times N}$, $B \in \mathbb{C}^{N \times 1}$, $C \in \mathbb{C}^{1 \times N}$, a linear RNN computes the following 1-D sequence-to-sequence map from an input $(u_0, \ldots, u_{L-1}) = u \in \mathbb{R}^L$ to output $(y_0, \ldots, y_{L-1}) = y \in \mathbb{C}^L$

$$
x_k = Ax_{k-1} + B \cdot u_k \quad , \quad y_k = C \cdot x_k.
\tag{7}
$$

In §A.3 we use a simple diagonalization argument to show if $A$ is diagonalizable over $\mathbb{C}$ then there exists an equivalent DLR of the same state size computing the same map. In particular, we show the following proposition, which asserts that DLRs are as expressive as general linear RNNs:

**Proposition 1.** *In Equation 7, let $A \in \mathbb{C}^{N \times N}$ be diagonalizable over $\mathbb{C}$ as $V\mathrm{diag}(\Lambda)V^{-1}$. Then, $\exists w \in \mathbb{C}^N$ such that DLR parameterized by $\Lambda, w$ (Equation 1) computes the same map as Equation 7.*

### 2.3 DLR Layer

In principle, one can directly parameterize our 1-D DLR map via $\Lambda, w \in \mathbb{C}^N$ and use Equation 4 to compute the output. Unfortunately, $||\Lambda||_\infty$ can become larger than 1 during training making the training unstable on long inputs as $K_{L-1}$ depends on terms as large as $\lambda_i^{L-1}$ which even for modest values of $L$ can be very large. Hence, we parameterize $\Lambda$ in log space and, following Goel et al. (2022), restrict the real parts to be negative. Our 1-D DLR map has parameters $(\log \Lambda)_{\text{re}}, (\log \Lambda)_{\text{im}} \in \mathbb{R}^N$, $w \in \mathbb{C}^{1 \times N}$. First, $\Lambda$ is computed as $\exp(-(\log \Lambda)_{\text{re}}^2 + i \cdot (\log \Lambda)_{\text{im}})$ where $i = \sqrt{-1}$ and the kernel is then computed via Equation 4.

Similar to S4, each DLR layer receives a sequence $u \in \mathbb{R}^{H \times L}$ of $H$-dimensional vectors and produces an output $y \in \mathbb{R}^{H \times L}$. The parameters of the layer are $(\log \Lambda)_{\text{re}}, (\log \Lambda)_{\text{im}} \in \mathbb{R}^N$ and $W \in \mathbb{C}^{H \times N}$. For each coordinate $h = 1, \ldots, H$, a kernel $K_h \in \mathbb{R}^L$ is computed as described above. The output $y_h \in \mathbb{R}^L$ for coordinate $h$ is computed from $u_h \in \mathbb{R}^L$ and $K_h$ using Equation 4. This is followed by a residual connection from $u$ to $y$. Moreover, to allow each output element to be a non-linear function of the input, a GELU non-linearity (Hendrycks & Gimpel, 2016) is applied and, finally, a position-wise linear projection $W_{\text{out}} \in \mathbb{R}^{H \times H}$ is then applied to enable information exchange among the $H$ coordinates. The DLR layer can be implemented in just a few lines of code (Figure 4).

**Initialization of DLR layer**   While the convolution view of DLR (Equation 4) scales efficiently on modern hardware to extremely long inputs, it is still a RNN and can suffer from vanishing gradients over a significant portion of the domain. Concretely, from Equation 3 we have $y_k = \sum_{j=0}^{k} K_j u_{k-j}$ and thus $\frac{\partial y_k}{\partial u_{k-j}} = K_j$. If $|K_{>c}| \ll 1$ then each $y_k$ would depend only on the local context $u_{k-c}, \ldots, u_k$. Furthermore, if $\max_i |\lambda_i| \ll 1$ then updating the values of $K_j$ for large values of $j$ will be slow since $K_j = \sum_{i=1}^{N} w_i \lambda_i^j$, $\frac{\partial K_j}{\partial \lambda_i} = w_i \lambda_i^{j-1} j$ and $\frac{\partial K_j}{\partial w_i} = \lambda_i^j$, which can hinder the ability to model long-range dependencies.

We parameterize the $\Lambda$ of DLR as $\exp(-(\log \Lambda)_{\text{re}}^2 + i \cdot (\log \Lambda)_{\text{im}})$ which is periodic in $(\log \Lambda)_{\text{im}}$ with period $2\pi$ and hence initialize $(\log \Lambda)_{\text{im}} \in \mathbb{R}^N$ as $(2\pi n/N)_{0 \leq n \leq N-1}$ by uniformly spacing it over its period. At this initialization with $(\log \Lambda)_{\text{re}} = \mathbf{0}$ and $L = N$ we would have $K = (\langle w, \Lambda^k \rangle)_{0 \leq k < L} = \text{DFT} \cdot w$. As DFT is invertible and well-conditioned, there is a $w$ for any arbitrary length-$N$ kernel and in particular one can express arbitrary long-range kernels.

Unless stated otherwise, each element of $(\log \Lambda)_{\text{re}}$ is initialized as $(e^r/2)^{1/2}$ where $r \sim \mathcal{U}(\log(.0005), \log(.5))$ to induce locality bias as $|\lambda_i| = \exp(-(\log \Lambda)_{\text{re,i}}^2) \leq 1$ is a decreasing function of $(\log \Lambda)_{\text{re,i}}$ and a smaller $|\lambda_i|$ leads to more local kernels. The real and imaginary parts of each element of $W$ are initialized from $\mathcal{N}(0, \sigma^2 = N^{-2})$. In all our experiments, the learning rate (and schedule) of all DLR parameters is same as that of other model parameters but weight decay is not applied to DLR parameters.

## 3   Experiments

Having proposed the conceptually simple DLR model, we now investigate the differences between DLR and different families of models including other state space models and attention. To this end, we start with synthetic tasks designed to pinpoint atomic capabilities (§3.1) and then turn to higher order long-range tasks involving hierarchical computation and reasoning over high-resolution synthetic images.

### 3.1   Atomic Tasks

We consider atomic tasks such as shifting, reversing, and sorting an input sequence to reveal any immediate shortcomings of a model and to see whether, from an expressivity point of view, one model is subsumed by another. Of particular interest to us will  the property that, unlike attention, convolutional models (*convnets*) apply the same manipulation to every input.

**Shift**   An input $x \in \mathbb{R}^L$ is sampled with elements from $\mathcal{N}(0,1)$ and normalized by $||x||_\infty$. For a parameter $C = 8$, the desired output $y \in \mathbb{R}^{L \times C}$ is $y_{ij} = x_{i - \frac{j \cdot L}{C}}$ for $j = 0, \ldots, C - 1$ and $x_{<0} = 0$, i.e. the output $y$ comprises uniformly-spaced right shifts of $x$. This task is similar to the "capacity task" proposed in (Voelker

et al., 2019). Note that given a sequence, convolving it with a one-hot kernel of the same length with 1 at position $r$, right shifts the sequence by $r$ positions.

**CumSum**   An input $x \in \mathbb{R}^L$ is sampled with elements from $\mathcal{N}(0, 1)$ and normalized by $||x||_\infty$. The output is $y \in \mathbb{R}^L$ with $y_i = (i+1)^{-1/2} \sum_{j \leqslant i} x_j$, where we scale by a factor of $(i+1)^{-1/2}$ as $n^{1/2}$ is standard deviation of the sum of $n$ standard gaussians. Note that convolving $x$ with the all-1's kernel of length $L$ produces $\sum_{j \leqslant i} x_j$.

**CumMax**   Same as CumSum, except the output $y \in \mathbb{R}^L$ is $y_i = \max_{j \leqslant i} x_j$. Unlike CumSum, this task cannot be described via a convolution with a fixed kernel.

**Reverse**   Same as CumSum, except that the output $y \in \mathbb{R}^L$ is $y_i = x_{L-1-i}$. In our experiments, to enable the use of unidirectional models on this task, we pad the input $x$ by $L$ zeros on the right to have an input of length $2L$ as such a model must observe the entire sequence $x$ before decoding. At the output, we consider the model prediction as the $L$ rightmost outputs.

**Sort**   Same as Reverse, except that in the output $y \in \mathbb{R}^L$, $y_i$ is the $i$'th closest element to $x_0$ i.e. we need to sort $x_i$'s according to their distance $|x_i - x_0|$ from the first element of the sequence.

**Select**   For a parameter $M = 32$, $x \in \mathbb{R}^{L+M}$ is sampled with elements from $\mathcal{N}(0, 1)$ and normalized by $||x||_\infty$. $M$ distinct positions $i_1 < \ldots < i_M$ are sampled from $0 \ldots L + M - 1$ and the output $y \in \mathbb{R}^M$ has $y_j = x_{i_j}$. In our experiments, we first pad $x$ with $M$ zeros on the right to get $x' \in \mathbb{R}^{L+2M}$ and form an input in $\mathbb{R}^{(L+2M) \times 2}$ by concatenating $0/1$ at each position indicating whether the position was among the $M$ selected positions. At the output, we consider the model prediction as the $M$ rightmost outputs.

**SelectFixed**   We also considered an easier variant of Select in which the randomly selected positions $i_1 < \ldots < i_M$ do *not* vary across samples, that is, the model has to copy the inputs from a fixed set of positions. This task is designed to investigate whether convnets like DLR having fixed kernels perform well on tasks where the desired sequence manipulation is context-independent.

**MIPS**   (maximum inner product search)  For a parameter $D = 4$, queries, keys and values $q, k, v \in \mathbb{R}^{L \times D}$ are sampled with elements from $\mathcal{N}(0, 1)$ and each vector is normalized by its euclidean norm. The output $y \in \mathbb{R}^{L \times D}$ is given by $y_i = v_{i'}$ where $i' = \text{argmax}_{j \leqslant i} \langle q_i, k_j \rangle$. An input in $\mathbb{R}^{L \times 3D}$ is formed by concatenating the corresponding query, key and value at each position. For the $i$'th query we do not consider the keys on it's right to enable the use of unidirectional models.

**Context-Shift**   $x \in \mathbb{R}^{L-2}$ is sampled with elements from $\mathcal{N}(0, 1)$ and normalized by $||x||_\infty$. A random shift value $s$ is sampled from $0, \ldots L - 2$ and the input $x' \in \mathbb{R}^L$ is formed as $x' = (\cos(2\pi s/L), \sin(2\pi s/L), x)$. The output $y \in \mathbb{R}^L$ is $y_i = x'_{i-s}$, where $x_{<0} = 0$. Unlike the Shift task, here the shift value is context-dependent as the model must infer $s$ to produce the output $y$.

**Solve**   For a given input length $L$, let $N$ be largest integer such that $L - N^2 - N \geqslant N$. A random orthonormal matrix $A \in \mathbb{R}^{N \times N}$ and a random unit vector $X \in \mathbb{R}^N$ are sampled and $B = AX$ is computed. An input $x \in \mathbb{R}^L$ is formed as $x = (a_1, b_1, \ldots, a_N, b_N, \mathbf{0}_{L-N^2-N}) \in \mathbb{R}^L$ where $a_i \in \mathbb{R}^N$ is the $i$'th row of $A$ and $b_i$ is the $i$'th element of $B$. The desired output $y \in \mathbb{R}^N$ is $X$. At the output, we consider the model prediction as the $N$ rightmost outputs. This task is inspired by recent works investigating whether Transformers can learn linear functions in-context (Garg et al., 2022).

**Solve-Fixed**   Same as Solve, except that the matrix $A$ is fixed and does not vary across the samples.

In all tasks, we include some rudimentary global positional information in the input $x \in \mathbb{R}^{T \times D}$ by modifying it as $x' \in \mathbb{R}^{T \times (D+2)}$ where $x'_i = (x_i, \cos(2\pi i/T), \sin(2\pi i/T))$.

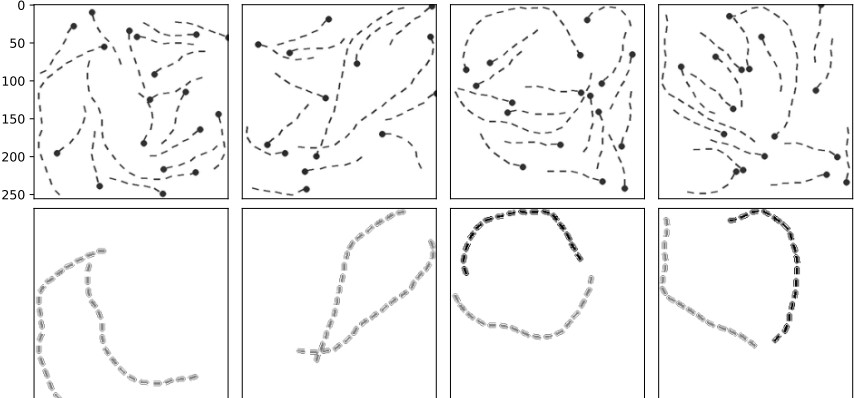

Figure 1: PATHFINDER-SEGMENTATION. For each input image in $\mathbb{R}^{M \times M}$ (top row) its corresponding label mask in $\{0, 1, 2\}^{M \times M}$ is shown at bottom. A pixel is labelled 0 (white color) if it does not lie on a main path, 2 (black) if it lies on a main path such that both ends of this path have black circles, and 1 (gray) otherwise. See §3.2 for details.

## 3.2 Higher-order Tasks

While the regression tasks in §3.1 are helpful at pinpointing atomic skills and weaknesses in model designs, we also devised classification tasks requiring multiple skills to approximate more realistic scenarios where we can pretrain on large amounts of data with strong supervision. In particular, we devised sequence-tagging versions of the two most challenging tasks in LRA, viz., LISTOPS and PATH-X that require hierarchical computation and detecting complex long-range spatial relationships in cluttered scenes. We convert these tasks to a sequence tagging format as the original classification setup provides a very sparse signal for training. Conversely, modern self-supervised models are based on rich supervision provided through a denoising objective. We argue that sequence tagging provides this dense supervision and is thus better-aligned with current training practices (El-Nouby et al., 2021; He et al., 2022; Krishna et al., 2022).

**ListOps-SubTrees**   We constructed a sequence-tagging version of the LISTOPS 10-way classification task where given a bracketed mathematical expression one has to compute its value in the set $\{0, \ldots, 9\}$. Unlike LISTOPS, instead of predicting just the value of the full expression, we tag each closing bracket with the value of the sub-expression computed at the corresponding sub-tree. For example, for the input expression "`[MAX 2 6 [MED [SM 3 1 6 ] 8 3 ] 4 5 ]`" the corresponding output is "`- - - - - - - - 0 - - 3 - - 6`" where the "`-`" labels are ignored and `[SM` denotes the sum modulo 10 operation. The input expressions were ensured to have lengths between 7000 and 8192 resulting in a median length $7.5\times$ larger than that of the LISTOPS data provided in LRA. The longer inputs make the task challenging from the perspective of long-range reasoning and supervision for every node of the expression tree provides a stronger training signal to the model making the task ideal for investigating the expressivity of long-range models.

**Pathfinder-Segmentation**   Finally, we constructed a sequence-tagging version of the original PATHFINDER task of (Linsley et al., 2018; Kim et al., 2020) where instead of predicting a single label for an image we predict a label for each pixel. Similar to PATHFINDER, a synthetic image is sampled in $\mathbb{R}^{M \times M}$ containing two long paths each with a preset number of dashes as shown in Figure 1. Along with these two main paths, several short "distractors" are also included. The task requires predicting a label for each pixel where the label of a pixel is 0 if it does not lie on a main path, 2 if it lies on a main path such that both ends of this path have black circles, and 1 otherwise.

Similar to LRA, we flatten the input image (and the output label mask) into a sequence of length $L = M^2$. Despite being continuous in the 2D image, a segment can potentially split across tens of thousands of positions in the flattened sequence. We generated data with $M = 256$ and $M = 512$ resulting in sequences of lengths $65K$ and $262K$ respectively, which is $4\times$ and $16\times$ longer compared to the PATH-X task from LRA making

Table 1: (Top) Average validation $R^2$ across batches on tasks described in §3.1, (Bottom) Relative time per step for models using the same input length. Actual input length for REVERSE and SORT is $2L$ and thus runtime is reported separately for them. Models with input lengths $2^{12}$, $2^9$ are trained for $40K$, $11K$ steps respectively.

| | DLR | $\text{DSS}_{\text{EXP}}$ | ATTENTION | DLR | DLR | ATTENTION |
|---|---|---|---|---|---|---|
| number of layers | 1 | 1 | 1 | 6 | 6 | 2 |
| params | $1.1M$ | $1.1M$ | $83K$ | $6.4M$ | $6.4M$ | $166K$ |
| $L$ | $2^{12}$ | $2^{12}$ | $2^{12}$ | $2^{12}$ | $2^9$ | $2^9$ |
| SHIFT | **1** | **.99** | .72 | **1** | 1 | 1 |
| CUMSUM | **1** | **1** | **1** | **1** | 1 | 1 |
| CUMMAX | .52 | .51 | **1** | **1** | 1 | 1 |
| SELECT-FIXED | **.97** | 0 | .72 | **1** | 1 | .94 |
| SOLVE-FIXED | **1** | .01 | 0 | **1** | 1 | .95 |
| REVERSE | .01 | 0 | .03 | **.99** | .95 | .28 |
| SOLVE | 0 | 0 | 0 | 0 | .95 | 0 |
| SELECT | 0 | 0 | .17 | 0 | .86 | .97 |
| SORT | 0 | 0 | 0 | .49 | .50 | .51 |
| CONTEXTSHIFT | 0 | 0 | 0 | .02 | .11 | .04 |
| MIPS | 0 | 0 | **.90** | 0 | .01 | .97 |
| SHIFT,CUMSUM, CONTEXTSHIFT,SELECT | $1\times$ | $1\times$ | $6.3\times$ | $5\times$ | $1\times$ | $1.1\times$ |
| REVERSE,SORT | $1\times$ | $1\times$ | $13.3\times$ | $5.4\times$ | $1\times$ | $1.8\times$ |

our setting considerably more challenging. Secondly, our task provides a stronger training signal to the model, again making it ideal for a study on expressivity of long range models.

For both of the above tasks, we generated $100K$ samples and used a $96/2/2$ train/validation/test split.

## 3.3 Results on Atomic Tasks

**Experimental setup** To compare attention-based contextualization to SSMs, we trained single DLR, $\text{DSS}_{\text{EXP}}$ and ATTENTION layers on the atomic tasks described in §3.1. We chose $\text{DSS}_{\text{EXP}}$ as a representative for models based on discretization of a continuous diagonal state space, since it was shown to be as performant as other S4-like models (Gupta et al., 2022; Gu et al., 2022b). To avoid any noise from non-contexualization layers such as feed-forward layers we formed an ATTENTION block by simply replacing the DLR contextualization in the DLR layer with a 4-head attention layer and rotary positional embeddings (Su et al., 2021). All models trained on atomic tasks are unidirectional and are trained for an equal number of steps with MSE loss, using a fresh batch sampled for every training and evaluation step. All experiments in §3.3 were performed on a single NVIDIA 3090 (24GiB) GPU.

Since atomic tasks are regression problems, we evaluate performance with $R^2$. At each evaluation step, we compute the $R^2$ score as $1 - \text{MSE}(y_{\text{pred}}, y_{\text{true}})/\text{MSE}(\text{mean}(y_{\text{true}}), y_{\text{true}})$ where $y_{\text{pred}}$ are model predictions, $y_{\text{true}}$ are true labels and $\text{mean}(y_{\text{true}}) \in \mathbb{R}$ is computed over the entire batch. We report the $R^2$ score averaged over a fixed number of evaluation steps. See §A.6 for training details.

**High-level overview** Tables 1 and 2 show the results for all models on different tasks and input lengths, based on which we can draw the following high-level conclusions. Firstly, in all experiments DLR performs as good as or better than $\text{DSS}_{\text{EXP}}$, illusutrating its viability as a long-range model. Secondly, we see in Table 2 that SSMs can scale to lengths that are infeasible for attention. For example, both $\text{DSS}_{\text{EXP}}$ and DLR get high results on SHIFT for sequences of length as large as $16K$ (and even beyond, see discussion below). To allow comparison between various models, we experiment in a setup where ATTENTION is feasible, namely for sequences of length 4096 and 512. We find that all convnet layers struggle on tasks where the desired sequence manipulation is context-dependent and, finally, that deep SSMs do not subsume attention. We now discuss these points in detail.

Table 2: Average validation $R^2$ across batches of a single layer on SHIFT task for input length $L$. Batch size = 4, hidden size = 32. All models use $N = 4096$ except where noted. DLR versions and $DSS_{EXP}$ use constant learning rate of 1e-5 and 1e-3 respectively. DLR-$\mathbb{R}$ denotes DLR with real-valued $\Lambda, w$.

| | | | $L$ | | | |
|---|---|---|---|---|---|---|
| model | params | $2^8$ | $2^{14}$ | $2^{16}$ | $2^{18}$ | $2^{20}$ |
| $DSS_{EXP}$ | 272$K$ | | .71 | .34 | .22 | .22 |
| DLR | 272$K$ | | .83 | .39 | .25 | .22 |
| SGCONV | 920$K$ | | .88 | .45 | .27 | .22 |
| DLR-prod | 272$K$ | | **1** | **1** | **.98** | **.78** |
| DLR, $N = 512$ | 35$K$ | | .30 | .22 | .22 | .22 |
| DLR-$\mathbb{R}$ | 137$K$ | .23 | | | | |

**Convnet layers struggle with context-dependent operations** As summarized in Table 1, SSM layers (DLR, $DSS_{EXP}$) are highly effective on tasks like SHIFT and CUMSUM that can be described via a small number of convolutional kernels. On the other hand, they fail on tasks such a REVERSE that potentially require a *large number of kernels* (e.g., a value at position $i$ needs to be right-shifted by $2L - i$ positions). Similarly, they fail on SELECT where, even though for each sample just $M = 32$ shift kernels should suffice, the value of these shifts is *context-dependent*, i.e., the kernels that need to be applied to an input vary with the input itself. By their very design, SSMs such as DLR learn context-independent kernels and hence struggle on such tasks. This hypothesis is further supported by the perfect performance of DLR on SELECT-FIXED in which the value of the shifts does not change with the inputs and hence this task can be described via a small number of convolutional kernels. Similarly, SSMs fail on CONTEXTSHIFT which is a context-dependent version of SHIFT.

**Compute-matched setting** Table 1 reveals that there are tasks such as CUMMAX and MIPS on which a single ATTENTION layer is more expressive than a single SSM layer. On the other hand, already for lengths as short as 4096, it is more than 6× slower. This raises the question that, instead of comparing a single DLR layer to an ATTENTION layer, what if we stack multiple DLR layers to the point that it takes similar time as an ATTENTION layer? After repeating the experiments with a 6-layer DLR model we indeed find almost perfect results on tasks such as REVERSE which require a large number of (context independent) kernels.

Interestingly, on context dependent tasks such as SORT, SELECT, MIPS, CONTEXTSHIFT and SOLVE even the deeper DLR model fails, demonstrating that ATTENTION is not subsumed by a deeper DLR stack and that further research is required to alleviate the shortcomings of SSMs on context dependent tasks. This is inline with Mehta et al. (2023) who reported a significant reduction in the perplexity of their SSM after sparingly interleaving in chunked attention layers, therefore suggesting that SSMs and attention offer complementary benefits.

We also experimented with the ATTENTION layer with additional feed-forward layers to match the parameter count of DLR but did not see improved results (§A.5).

**Scalability on Shift task** Having established the encouraging performance of DLR layers on tasks such as SHIFT that require only a few kernels, we explore if this performance holds while scaling the input length to larger values and whether DLR can learn extremely long kernels with high resolution. As shown in Table 2, even a single DLR layer provides excellent performance on this task on $16K$-long sequences, but the performance starts to degrade beyond that. Interestingly, we found that the DLR-prod version of DLR (§2.1) performs significantly better with nearly perfect performance on lengths as large as $1M$. This suggests that there is room for more complex kernel designs having better performance on certain tasks. We note that using a large enough $N$ is essential for expressing long range kernels with high-resolution and that an inadequately small[2] $N$ leads to a reduced performance (Table 2).

---

[2] We note that $C = 8$ one-hot kernels, i.e., model dimension $H = 8$, should suffice on SHIFT and it is the state size $N$ that needs to be large. In practice, a large $N$ can lead to high memory usage with models like S5 (Smith et al., 2023) that work directly at state level (Equation 2) with a $\Omega(B \cdot N \cdot (H + L \log L))$ complexity.

Table 3: (left) Token-wise accuracy on test set of LISTOPS-SUBTREES. (right) macro F1-score / macro accuracy on test set of PATHFINDER-SEGMENTATION where we compute the F1-score/accuracy individually for each label class and average the 3 values. ✗ denotes the experiment was infeasible due to compute constraints. Relative time per step is in parenthesis. See §3.4 for more details.

| | LISTOPS-SUBTREES | PATHFINDER-SEGMENTATION | | |
| | | $128 \times 128$ | $256 \times 256$ | $512 \times 512$ |
| sequence length | $8K$ | $16K$ | $65K$ | $262K$ |
| number of layers | 6 | 5 | 6 | 12 |
| LOCALATTENTION | **94.0** (8×) | 95.2 / 98.4 (19×) | ✗ | ✗ |
| DSS$_{\text{EXP}}$ | 83.8 (1×) | 94.4 / 97.2 (1×) | 89.4 / 96.2 | 60.4 / 79.3 |
| DLR | 85.7 (1×) | **96.8** / 97.7 (1×) | **94.0** / 96.3 | **76.3** / 92.8 |

**Restricting DLR parameters to reals** We also experimented with a version of DLR denoted as DLR-$\mathbb{R}$ in which we restrict $\Lambda, w$ in Equation 1 to be real-valued and form $\Lambda = \exp(-(\log \Lambda)_{\text{re}}^2)$ in §2.3. While this version managed to give an $R^2$ score of 1 on CUMSUM, it scored 0 on SELECT-FIXED and, as shown in Table 2, failed on SHIFT with lengths as short as 256. This suggests that methods such as EMA (Ma et al., 2023), based on diagonal state spaces with purely-real parameterizations are incapable of modeling even short arbitrary kernels and we formally prove this in §A.4. This further suggests that long-range interactions in a multi-layer MEGA stack are captured by chunked attention, with EMA potentially providing some inter-chunk communication due to the use of non-overlapping chunks.

### 3.4 Results on Higher-order Tasks

The experiments and analysis in §3.3 give us new insights into the workings of SSMs and their limitations. We would now like to understand how prohibitive these limitations are on the higher-order tasks defined in §3.2. To that end, we trained multi-layer models on these tasks and summarize the results in Table 3. As the input lengths of these tasks are infeasible for ATTENTION, we used a tractable version of the ATTENTION block defined in §3.3, denoted as LOCALATTENTION, where we chunk the input to the attention layer into non-overlapping chunks of length 1024 (or 4096 for image tasks) and allow each chunk to attend to itself and the adjacent chunk(s).

**ListOps-SubTrees** Similar to the experiments in §3.3, the performance of DLR is again slightly better than DSS$_{\text{EXP}}$. Interestingly, although LOCALATTENTION is 8× slower than SSMs, in terms of performance it outperforms SSMs, which contradicts the low historical performance of Transformer variants on the LISTOPS version from LRA, highlighting the benefits of a dense training signal and reaffirming the orthogonal benefits of attention and SSMs.

Secondly, while DLR delivers a per-token accuracy of 85.7, error analysis (Figure 2) reveals that this can mainly be attributed to shallow sub-expressions and that the model performs poorly on expressions with a parse tree of height more than 2, where the "length" of a path is the number of operators on it. For height beyond 3, the errors at the children compound, leading to errors higher up the tree. This suggests that sequence models including SSMs struggle at hierarchical computation.

Breaking down the performance with respect to the operator at a node in an expression tree reveals that most errors can be attributed to operators such as [SM (sum modulo 10) that are sensitive to *all* the inputs, i.e. perturbing the value of even a single argument will change the output of the operator. Hence, the model must compute all the sub-expressions correctly. On the other hand operators such as [MAX are more robust to small perturbations in their arguments making it harder for the errors at the children to propagate to the parent.

**Pathfinder-Segmentation** Unlike previous experiments, we trained bidirectional models on this task. Due to the high imbalance between the pixel-label classes 0/1/2, we report macro F1-score (and macro accuracy) where we compute the F1-score (accuracy) individually for each label class and average the 3 values. DLR not only outperforms the baselines but delivers an impressive performance for images as large as $256 \times 256$

Table 4: Sequence classification accuracy on Long Range Arena tasks and 10-way Speech Commands task. Performance of Transformer, S4D and S4 is as reported in (Gu et al., 2022b). Performance of S4 on Speech Commands is as reported in (Gu et al., 2022a). ✗ denotes chance performance or computationally infeasible.

|  | LISTOPS | TEXT | RETRIEVAL | PATH-X | SPEECHCOMMANDS |
|---|---|---|---|---|---|
| Transformer | 36.4 | 64.3 | 57.5 | ✗ | ✗ |
| S4D-Inv | 60.2 | **87.3** | **91.1** | 92.8 | |
| S4-LegS | 59.6 | 86.8 | 90.9 | **96.4** | **98.3** |
| DLR | **60.5** | 86.7 | 89.1 | 94.5 | 97.1 |

Table 5: Train/test cross-entropy loss on PG-19 text corpus. Text is tokenized using T5 tokenizer and chunked into sequences of size 4096. Transformer uses the hardware-optimized attention implementation in PyTorch 2.0 based on (Dao et al., 2022). Details in §A.6.

|  | experts | params | throughput | post-norm | pre-norm |
|---|---|---|---|---|---|
| Transformer | 1 | 36M | 1.1× | diverged | 2.88 / 2.90 |
| Transformer | 16 | 318M | 1.0× | diverged | 2.52 / **2.63** |
| DLR | 16 | 312M | 1.1× | 2.70 / **2.86** | 2.71 / 2.88 |

(input length $65K$), corroborated by the model predictions on random samples from the validation set (Figure 3, top). The $512 \times 512$ case with input length $262K$ is more challenging as each layer of the model needs to contextualize over tens of thousands of positions. While DLR outperforms the baselines, it does leave a significant room for improvement, and indeed as seen from the model predictions (Figure 3 bottom) it makes quite a few errors.

Compared to LISTOPS-SUBTREES, PATHFINDERSEGMENTATION requires contextualization over significantly longer ranges and in this case LOCALATTENTION with chunk size 4096 is outperformed by DLR both in terms of performance and speed. Due to the long training times we could not report its performance on the 256 and 512 cases. In the future, we plan on benchmarking other Transformer variants to conclusively determine if there is any benefit of using them over SSMs on long-range tasks. The large gap between DLR and $DSS_{EXP}$ in the $512 \times 512$ case can potentially be reduced with better hyperparameter tuning of $DSS_{EXP}$ and we leave this for future work. Training details are provided in §A.6.

### 3.5 Results on Sequence Classification and Language Modeling

**Long Range Arena**   We also benchmarked DLR on a subset of tasks from Long Range Arena as well as on Speech Commands raw speech classification (Warden, 2018). Unlike the previous tasks considered in this work, these are sequence classification tasks with sparse supervision (e.g. in case of PATH-X a single binary label per image). As shown in Table 4, we again find the performance of DLR to be on a par with that of the best performing SSMs S4 and S4D, in addition to having a much simpler and cleaner formulation.

**Language Modeling**   To demonstrate the effectiveness of DLR at modeling complex real-word tasks, we performed causal language modeling on the PG-19 text corpus consisting of English books (Rae et al., 2020). As shown in Table 5, we find the performance and throughput of DLR to be comparable to that of Transformers with hardware-optimized attention implementation (Dao et al., 2022) while enjoying a $O(L)$ complexity at decoding time compared to $O(L^2)$ complexity in case of Transformers. We also find DLR to be more robust to the placement of the layer-norm layers compared to Transformers which are known to be highly sensitive to their placement (Xiong et al., 2020).

## 4 Conclusion

In this work, we provide two contributions towards better understanding of state space models (SSMs) for modeling long sequences. First, we propose Diagonal Linear RNNs, which simplify diagonal state spaces

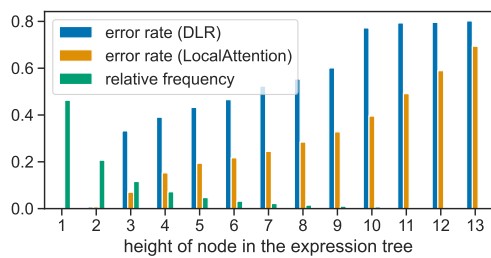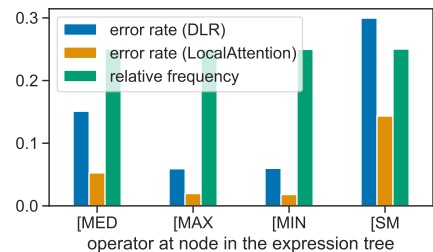

Figure 2: Breakdown of errors made by DLR and LOCALATTENTION on LISTOPS-SUBTREES validation set according to (left) height of the node, and (right) the operator at the node. See §3.4 for details.

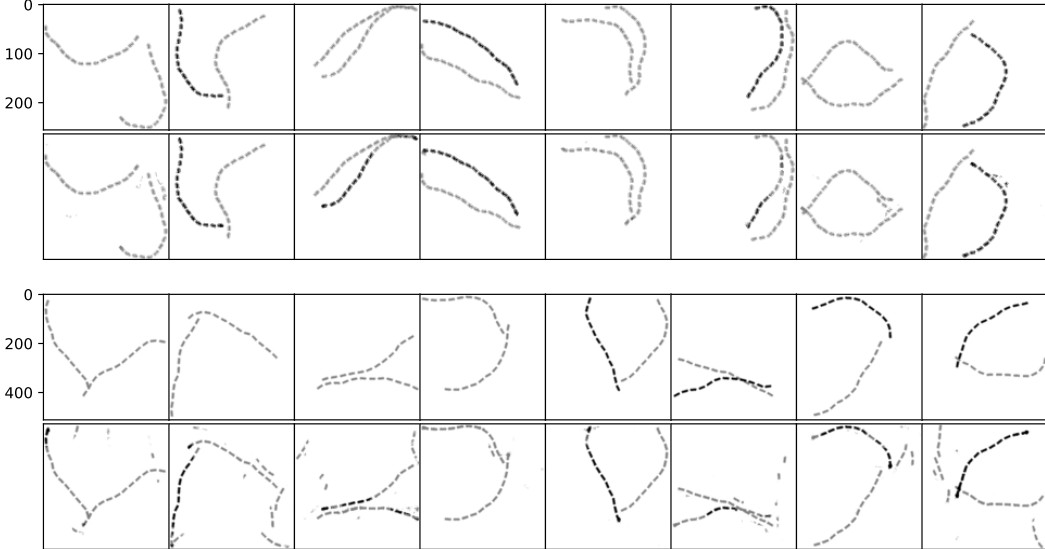

Figure 3: Predictions of DLR on random samples from the validation set of PATHFINDER-SEGMENTATION-256 (top) and PATHFINDER-SEGMENTATION-512 (bottom). In each image, the first row shows the gold label mask and the second row shows the model predictions. See §3.4 for details.

by dropping the continuous state space discretization, and then propose a suitable initialization scheme. We empirically show that DLR performs as well as or better than DSS on a wide range of synthetic tasks. Second, we provide insights onto the capabilities of SSMs by comparing them to attention-based models on both atomic tasks and high-order tasks. We show that SSMs excel at tasks that can be handled through a small number of convolution kernels, but struggle on context-dependent tasks, or where a large number of context-independent kernels are necessary. Our results offer insights that can steer future research, and our proposed synthetic benchmarks provide a rich test-bed for the research community.

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

# A Supplemental Material

## A.1 Fast convolution via FFT

For $u, K \in \mathbb{C}^L$ the Circular Convolution Theorem states that,

$$\text{circulant}(K) \cdot u \;=\; \begin{bmatrix} K_0 & K_{L-1} & \cdots & K_1 \\ K_1 & K_0 & \ddots & \vdots \\ \vdots & \ddots & \ddots & K_{L-1} \\ K_{L-1} & \cdots & K_1 & K_0 \end{bmatrix} \cdot u \;=\; \text{invFFT}_L(\text{FFT}_L(K) * \text{FFT}_L(u)).$$

where $*$ denotes elementwise multiplication. As $\text{FFT}, \text{invFFT}$ can be done in $O(L \log L)$ time this provides a fast algorithm for circulant matrix-vector product (Cormen et al., 2009). In practice, linear systems can often be expressed as a circulant matrix-vector product and is also true in the case of Equation 4 which can be equivalently expressed as

$$[y_0 \;\ldots\; y_{L-1} \mid \ldots] \;=\; \text{circulant}([K \mid 0 \;\ldots\; 0])_{2L \times 2L} \cdot [u_0 \;\ldots\; u_{L-1} \mid 0 \;\ldots\; 0]_{2L \times 1}.$$

Similarly, Equation 6 can be expressed as a standard Toeplitz matrix-vector product

$$y \;=\; \begin{bmatrix} \vec{K}_0 & \cev{K}_0 & \cdots & \cev{K}_{L-2} \\ \vec{K}_1 & \vec{K}_0 & \ddots & \vdots \\ \vdots & \ddots & \ddots & \cev{K}_0 \\ \vec{K}_{L-1} & \cdots & \vec{K}_1 & \vec{K}_0 \end{bmatrix} \cdot u$$

which can be expressed as a circulant matrix-vector product of size $2L$ as

$$[y_0 \;\ldots\; y_{L-1} \mid \ldots] \;=\; \text{circulant}([\vec{K}_0, \ldots, \vec{K}_{L-1}, X, \cev{K}_{L-2}, \ldots, \cev{K}_0]) \cdot [u_0 \;\ldots\; u_{L-1} \mid 0 \;\ldots\; 0]_{2L \times 1}$$

where $X$ is allowed to be any value.

## A.2 DLR-prod is a DLR

Equations 3 and 4 imply that for a DLR parameterized by $\Lambda = (\lambda_i)_{1 \leqslant i \leqslant N}$ and $w = (w_i)_{1 \leqslant i \leqslant N}$

$$y_k \;=\; \sum_{j=0}^{k} \underbrace{\left( \sum_{i=1}^{N} w_i \lambda_i^j \right)}_{K_j} u_{k-j}$$

where $K_k = \sum_{i=1}^{N} w_i \lambda_i^k \in \mathbb{C}$. Then,

$$\begin{aligned}
\text{Re}(K_k) \cdot \text{Im}(K_k) \in \mathbb{R} &= \left( \sum_{m=1}^{N} \text{Re}(w_m \lambda_m^k) \right) \left( \sum_{n=1}^{N} \text{Im}(w_n \lambda_n^k) \right) \\
&= \sum_{m=1}^{N} \sum_{n=1}^{N} \text{Re}(w_m \lambda_m^k) \text{Im}(w_n \lambda_n^k) = \sum_{m,n} \frac{(w_m \lambda_m^k + \overline{w}_m \overline{\lambda}_m^k)}{2} \frac{(w_n \lambda_n^k - \overline{w}_n \overline{\lambda}_n^k)}{2i} \\
&= -i/4 \sum_{m,n} w_m w_n (\lambda_m \lambda_n)^k - w_m \overline{w}_n (\lambda_m \overline{\lambda}_n)^k + \overline{w}_m w_n (\overline{\lambda}_m \lambda_n)^k - \overline{w}_m \overline{w}_n (\overline{\lambda}_m \overline{\lambda}_n)^k \\
&= \sum_{j=1}^{4N^2} \tilde{w}_j \tilde{\lambda}_j^k
\end{aligned}$$

where clearly $\tilde{w}_j$ and $\tilde{\lambda}_j$ can be appropriately defined from the expression above it. Finally, $\sum_{j=1}^{4N^2} \tilde{w}_j \tilde{\lambda}_j^k$ is the expression of a DLR kernel of size at most $4N^2$.

**Kronecker product of DLR's** In general, given two DLRs parameterized by $\Lambda, w \in \mathbb{C}^M$ and $\tilde{\Lambda}, \tilde{w} \in \mathbb{C}^N$ we define their *Kronecker product* as the DLR parameterized by $\Lambda \otimes \tilde{\Lambda}, w \otimes \tilde{w} \in \mathbb{C}^{MN}$. It is easy to see that the kernel $K(\Lambda \otimes \tilde{\Lambda}, w \otimes \tilde{w}) \in \mathbb{C}^L$ of this DLR is the elementwise product of kernels $K(\Lambda, w)$ and $K(\tilde{\Lambda}, \tilde{w})$ as

$$K(\Lambda, w)_k \cdot K(\tilde{\Lambda}, \tilde{w})_k \in \mathbb{C} = \left( \sum_{m=1}^{M} w_m \lambda_m^k \right) \left( \sum_{n=1}^{N} \tilde{w}_n \tilde{\lambda}_n^k \right)$$

$$= \sum_{m=1}^{N} \sum_{n=1}^{N} (w_m \tilde{w}_n)(\lambda_m \tilde{\lambda}_n)^k = K(\Lambda \otimes \tilde{\Lambda}, w \otimes \tilde{w})_k.$$

## A.3 DLR vs Linear RNN

As stated in §2.2, for $A \in \mathbb{C}^{N \times N}$, $B \in \mathbb{C}^{N \times 1}$, $C \in \mathbb{C}^{1 \times N}$, a linear RNN computes the following 1-D sequence-to-sequence map from an input $(u_0, \ldots, u_{L-1}) = u \in \mathbb{R}^L$ to output $(y_0, \ldots, y_{L-1}) = y \in \mathbb{C}^L$ via the recurrence

$$x_k = A x_{k-1} + B \cdot u_k \quad , \quad y_k = C \cdot x_k.$$

**Proposition.** *In the above equation, let $A \in \mathbb{C}^{N \times N}$ be diagonalizable over $\mathbb{C}$ as $V \mathrm{diag}(\Lambda) V^{-1}$. Then, $\exists w \in \mathbb{C}^N$ such that DLR parameterized by $\Lambda, w$ (Equation 1) computes the same map as the above linear RNN.*

*Proof.* Assuming $x_{-1} = 0$, the linear RNN can be unrolled as

$$y_k = \sum_{j=0}^{k} C A^j B u_{k-j} = \sum_{j=0}^{k} C \left( V \mathrm{diag}(\Lambda) V^{-1} \right)^j B u_{k-j}$$

$$= \sum_{j=0}^{k} C V \left( \mathrm{diag}(\Lambda) \right)^j V^{-1} B u_{k-j} = \sum_{j=0}^{k} (CV) \mathrm{diag}(\Lambda^j)(V^{-1}B) u_{k-j}$$

Let $CV \in \mathbb{C}^{1 \times N} = (c_1, \ldots, c_N)$, $V^{-1}B \in \mathbb{C}^{N \times 1} = (b_1, \ldots, b_N)$, $w = (c_1 b_1, \ldots, c_N b_N)$ and $\Lambda = (\lambda_1, \ldots, \lambda_N)$. Then,

$$y_k = \sum_{j=0}^{k} (CV) \mathrm{diag}(\Lambda^j)(V^{-1}B) u_{k-j} = \sum_{j=0}^{k} \sum_{i=1}^{N} c_i \lambda_i^j b_i u_{k-j} = \sum_{j=0}^{k} \sum_{i=1}^{N} w_i \lambda_i^j u_{k-j}$$

which is identical to the expression of $y_k$ in Equation 3 of a DLR parameterized by $\Lambda, w$. $\square$

## A.4 On the expressivity of DLR-$\mathbb{R}$

In DLR-$\mathbb{R}$, the $\Lambda, w$ in Equation 1 are restricted as $\Lambda \in (0, 1]^N$ and $w \in \mathbb{R}^N$. The kernel $K \in \mathbb{R}^L$ in Equation 4 can be written as a Vandermonde matrix-vector product $K = w P_{N \times L}$ where $P_{ij} = \lambda_i^j$. It is known that if $\Lambda \in \mathbb{R}_+^N$, $P$ is highly ill-conditioned (Gautschi, 2020; Aubel & Bölcskei, 2019). Here we explain its failure on the SHIFT task (Table 2) and in Claim 1 prove that the norm of the solution grows exponentially with $N$. Shifting an input $u \in \mathbb{R}^L$ by $S$ positions requires a one-hot kernel $\mathbf{1}_S \in \mathbb{R}^L$ that is 1 at position $S$. Assuming, $S = L = N$, we have $K = \mathbf{1}_N$ and need to solve for $w \in \mathbb{R}^N$ such that $wP = \mathbf{1}_N$.

**Claim 1.** *Let $\mathbf{1}_N \in \mathbb{R}^{1 \times N}$ denote the one-hot vector with 1 at position $N$. Let $P \in \mathbb{R}^{N \times N}$ with $P_{ij} = \lambda_i^j$ be a $N \times N$ Vandermonde matrix with each $\lambda_i \in [0, 1]$. If $\mathbf{1}_N = wP$ for $w \in \mathbb{R}^N$, then $||w||_\infty \geqslant 2^{2N - \tilde{O}(\log N)}$.*

*Proof.* We first claim that one necessarily requires $N$ distinct $\lambda_i$'s. If not then let $\lambda_1, \ldots, \lambda_r$, $r < N$ be distinct and $w_{1 \times r} P_{r \times N} = \mathbf{1}_N$. The first $r$ equations can be written as $w_{1 \times r} P_{r \times r} = \mathbf{0}_r$ where $P_{ij} = \lambda_i^j$, $1 \leqslant i \leqslant r$, $j < r$. As $\lambda_1, \ldots, \lambda_r$ are assumed to be distinct, $P_{r \times r}$ is invertible and hence $w_{1 \times r} = \mathbf{0}$ which does not satisfy $w_{1 \times r} P_{r \times N} = \mathbf{1}_N$. Therefore, $\lambda_1, \ldots, \lambda_N$ must be distinct which implies $P$ is invertible and $w = \mathbf{1}_N P^{-1}$. By the expression of the inverse of a Vandermonde, this gives the unique solution

Table 6: (Top) Average validation $R^2$ across batches on tasks described in §3.1, (Bottom) Relative time per step for models using the same input length. Actual input length for Reverse and Sort is $2L$. Models with input lengths $2^{12}$, $2^9$ are trained for $40K$, $11K$ steps respectively.

| | DLR | Attention + 2×FF | Attention | DLR | DLR | Attention | Attention + 2×FF |
|---|---|---|---|---|---|---|---|
| number of layers | 1 | 1 | 1 | 6 | 6 | 2 | 2 |
| params | 1.1M | 1.1M | 83K | 6.4M | 6.4M | 166K | 2.2M |
| $L$ | $2^{12}$ | $2^{12}$ | $2^{12}$ | $2^{12}$ | $2^9$ | $2^9$ | $2^9$ |
| Shift | **1** | .69 | .72 | **1** | 1 | 1 | 1 |
| Select-Fixed | **.97** | 0 | .72 | **1** | 1 | .94 | 1 |
| Solve-Fixed | **1** | 0 | 0 | **1** | 1 | .95 | .96 |
| Reverse | .01 | .04 | .03 | **.99** | .95 | .28 | .32 |
| Solve | 0 | 0 | 0 | 0 | .95 | 0 | 0 |
| Select | 0 | 0 | .17 | 0 | .86 | .97 | .64 |
| Sort | 0 | 0 | 0 | .49 | .50 | .51 | .51 |
| ContextShift | 0 | 0 | 0 | .02 | .11 | .04 | .05 |
| Shift | 1× | 7× | 6.3× | 5× | 1× | 1.1× | 1.7× |
| Reverse,Sort | 1× | 14.3× | 13.3× | 5.4× | 1× | 1.8× | 2.6× |

$w_i = \frac{-1}{\prod_{j \neq i}(\lambda_j - \lambda_i)}$, $1 \leqslant i \leqslant N$. Clearly, to show the lower bound for $||w||_\infty$, it suffices to show it for $(\prod_i |w_i|)^{1/N}$. We have $(\prod_i |w_i|)^{1/N} = (\prod_{i<j} |\lambda_j - \lambda_i|)^{-2/N}$. If each $\lambda_i \in [0,1]$, it can be shown that the Vandermonde determinant $\prod_{i<j} |\lambda_j - \lambda_i| \leqslant (c + o(1))2^{-N^2}\sqrt{(N-1)!}(8e)^{N/2}N^{3/8} = 2^{-N^2+O(N\log N)}$ for some fixed $c > 0$ (Pinelis) and hence $(\prod_i |w_i|)^{1/N} \geqslant 2^{2N-O(\log N)}$. $\qquad\square$

## A.5 Additional Experiments

In Table 1, a single Attention layer uses fewer parameters than a DLR layer. For a comparison in a setting where both use same the number of parameters, we repeated the experiments with the Attention layer additionally followed by two feed-forward layers each with a feed-forward dimension of 2048. The results are presented in Table 6.

## A.6 Experimental Setup

In this section, we describe the training details for the experiments presented in §3. Our experimental setup was built on top of an earlier version of the training framework provided by the S4 authors[3] and our implementations of DLR, $\text{DSS}_{\text{EXP}}$ and Attention leverage the PyKeOps library for memory efficiency (Charlier et al., 2021).

**Details for Table 1** Input $x \in \mathbb{R}^{L \times D}$ is linearly projected to $xW \in \mathbb{R}^{L \times d}$ where $d$ is the model dimension. For a desired output $y \in \mathbb{R}^{L' \times D'}$ we take the output $o \in \mathbb{R}^{L \times d}$ of the model and linearly project it to $oW' \in \mathbb{R}^{L \times D'}$ and take its rightmost $L'$ positions as the prediction. All experiments use post-norm Layer Normalization. Model dimension $H = 128$ and weight decay of optimizer was 0. For SSMs, state size $N = 4096$. Learning rate (and schedule) of SSM layer parameters was same as other model parameters. Constant learning rate schedule was used. In $\text{DSS}_{\text{EXP}}$, $\log \Lambda$ is initialized as $(-.5 + 2\pi i n)_{0 \leqslant n \leqslant N-1}$ (Gu et al., 2022b). Hyperparameters are provided in Table 7.

**Details for Table 2** Same as details for Table 1 except $H = 32$, batch size is 4, number of layers is 1 and number of steps was $86K$. Learning rate was 1e-5 for DLR, 1e-3 for $\text{DSS}_{\text{EXP}}$ and 1e-5 for SGConv (Li et al., 2023). $\alpha$-min and $\alpha$-max were 1e-5 for SGConv to avoid signal decay, kernel dimension $d = 4096$ and number of concatenated kernels (i.e. number of scales) was computed so that the resulting kernel is at least as long as the input. The kernel parameters were initialized from $\mathcal{N}(0, \sigma^2 = d^{-2})$.

Experiments in Table 1 and 2 were performed on a single NVIDIA 3090 (24GiB).

---

[3] https://github.com/HazyResearch/state-spaces

Table 7: Hyperparameters for Table 1 on all tasks except MIPS. Exceptions are detailed in §A.6. LR is initial learning rate.

|  | L | layers | H | N | dt-min | dt-max | LR | Batch Size | steps | epochs |
|---|---|---|---|---|---|---|---|---|---|---|
| DLR | $2^{12}$ | 1 / 6 | $2^7$ | $2^{12}$ | 1e-5 | 1e-5 | 1e-4 | 16 | $40K$ | 12 |
| $DSS_{EXP}$ | $2^{12}$ | 1 | $2^7$ | $2^{12}$ | 1e-4 | 1e-2 | 1e-3 | 16 | $40K$ | 12 |
| ATTENTION | $2^{12}$ | 1 | $2^7$ |  |  |  | 1e-3 | 16 | $40K$ | 12 |
| DLR | $2^9$ | 6 | $2^7$ | $2^{12}$ | 1e-5 | 1e-5 | 5e-5 | 64 | $11K$ | 12 |
| ATTENTION | $2^9$ | 2 | $2^7$ |  |  |  | 1e-3 | 64 | $11K$ | 12 |

**Details for Table 3**  In all models and tasks, after each model block, a GLU non-linearity (Dauphin et al., 2017) was additionally applied. Cosine learning rate schedule with linear warmup was used. The test metrics were measured at the checkpoint with the highest validation accuracy. Each metric was computed for an individual test batch and averaged across the batches in the test set, which depending upon the metric might vary with the batch size. SSM trainings on LISTOPS-SUBTREES and PATHFINDER-SEGMENTATION-128 were performed on single 3090, whereas for the 256, 512 cases we used 3 3090's and 7 V100's respectively.

Table 8: Hyperparameters for DLR models in Table 3. LS denotes LISTOPS-SUBTREES and PS denotes PATHFINDER-SEGMENTATION. Exceptions are detailed in §A.6. WD is weight decay, B is batch size. For efficiency, instead of constructing kernels of length equal to the input length, they are restricted to "kernel size".

|  | L | layers | H | N | dt-min | dt-max | LR | B | steps | epochs | WD | kernel size |
|---|---|---|---|---|---|---|---|---|---|---|---|---|
| LS | $2^{13}$ | 6 | $2^7$ | $2^{10}$ | 1e-4 | 1e-1 | 8e-4 | 32 | $300K$ | 100 | 0.01 | $2^{13}$ |
| PS | $2^{14}$ | 5 | $2^7$ | $2^{10}$ | 1e-4 | 1e-1 | 1e-4 | 16 | $150K$ | 30 | 0 | $2^{13}$ |
| PS | $2^{16}$ | 6 | $2^7$ | $2^{10}$ | 1e-4 | 1e-1 | 5e-5 | 18 | $178K$ | 40 | 0 | $2^{15}$ |
| PS | $2^{18}$ | 12 | $2^6$ | $2^{11}$ | 1e-4 | 1e-1 | 1e-5 | 14 | $114K$ | 21 | 0 | $2^{15}$ |

**Details for Table 4**  Same as the details for Table 3 as listed above with the following changes.

Table 9: Hyperparameters for bidirectional DLR model in Table 4. Layer norm is used and max pooling is applied at the model output to form a single vector representation. Dropout is traditional per-element dropout.

|  | L | layers | H | N | dt-min | dt-max | LR | B | steps | epochs | WD | kernel size | dropout |
|---|---|---|---|---|---|---|---|---|---|---|---|---|---|
| PATH-X | $2^{14}$ | 6 | $2^8$ | $2^{11}$ | 1e-4 | 1e-1 | 1e-4 | 16 | $500K$ | 50 | 0.05 | $2^{14}$ | 0 |
| TEXT | $2^{11}$ | 4 | $2^7$ | $2^8$ | 0.2 | 0.5 | 1e-4 | 128 |  | 200 | 0.0 | $2^{11}$ | 0.3 |
| RETRIEVAL | $4K$ | 6 | $2^8$ | $2^8$ | 0.05 | 0.5 | 1e-4 | 32 | $150K$ | 30 | 0.05 | $4K$ | 0.05 |
| LISTOPS | $2K$ | 6 | $2^7$ | $2^{10}$ | 1e-3 | 0.5 | 4e-4 | 50 | $320K$ | 160 | 0.05 | $2K$ | 0.2 |
| SPEECHCOMMANDS | $16K$ | 6 | $2^7$ | $2^{11}$ | 1e-3 | 0.5 | 1e-4 | 20 |  | 200 | 0.0 | $2K$ | 0.05 |

**Details for Table 5**  We tokenized the texts using T5 tokenizer and concatenated them to get a long sequence of tokens. This was chunked into sequences of size 4096. Each block consists of a linear DLR layer (without non-linearity or output projection) followed by a heterogeneous Switch layer with 16 feed-forward experts (Fedus et al., 2021). Each expert independently processes 1/16'th of its top tokens. Their outputs are scaled by the router probability and summed, allowing a token to be processed by none/multiple experts. During evaluation, we doubled the expert capacity to 2/16. In pre-norm, layer norm is applied to inputs of each sub-layer whereas in post-norm it is applied to their outputs. In pre-norm, an additional layer-norm is applied to the model output. Embedding size is 128 and input-output embeddings are tied. Learning rate for DLR parameters was 2e-4 whereas for other parameters it was 1e-3. For Transformer, we replaced DLR part with attention with 8 heads of size 64 each. For simplicity, we did not use load-balancing loss for Switch layers and observed some overfitting. Each run was performed on 3 A100's 40GiB for 1 day. Throughput in Table 5 was measured on single A100 80GiB.

Table 10: Hyperparameters for unidirectional DLR model in Table 5.

| | L | layers | H | N | dt-min | dt-max | LR | B | steps | epochs | WD | kernel size | dropout |
|---|---|---|---|---|---|---|---|---|---|---|---|---|---|
| PG-19 | $2^{12}$ | 16 | 384 | $2^9$ | 1e-3 | 1e-1 | 1e-3 | 120 | $60K$ | 10 | 0.1 | $2^{12}$ | 0 |

```python
def dlr_kernel(L, prod=False):
    # L: kernel length
    # Lambda_log_re: [N],  Lambda_log_im: [N],  W: [H N 2]  (floats)
    Lambda_log_re, Lambda_log_im, W = get_layer_parameters()

    # convert reals to complex
    Lambda_log = -Lambda_log_re**2 + 1j*Lambda_log_im          # [N]
    W = W[...,0] + 1j*W[...,1]                                  # [H N]

    pos = torch.arange(L, device=W.device)                     # [L]
    P = (Lambda_log.unsqueeze(-1) * pos).exp()                 # [N L]
    K = W.matmul(P)                                            # [H L]
    return K.real * K.imag if prod else K.real                 # [H L]

def state_space(u, bidirectional=False):
    # u: batch of input sequences
    # B: batch size, H: hidden size, L: sequence length
    B, H, L = u.shape

    if not bidirectional:
        # compute state space kernel for each of H coordinates
        K = dlr_kernel(L)                                      # [H L]
    else:
        # compute two state space kernels for each coordinate
        # one for each direction
        K = dlr_kernel(L)                                      # [2H L]
        K = torch.cat((K[:H], K[H:].flip(dim=-1)), dim=-1)     # [H 2L]

    # circulant matrix-vector product of size 2L
    K_f = torch.fft.rfft(K, n=2*L)                             # [H L+1]
    u_f = torch.fft.rfft(u, n=2*L)                             # [B H L+1]
    y_f = K_f * u_f                                            # [B H L+1]
    y = torch.fft.irfft(y_f, n=2*L)[...,:L]                    # [B H L]

    # residual connection, non-linearity, output projection not shown
    return y
```

Figure 4: Core implementation of DLR contextualization (§2.1) in PyTorch.

