# OpenReview forum: "Simplifying and Understanding State Space Models with Diagonal Linear RNNs"
_TMLR — Rejected by TMLR_

### Review · Reviewer_jUMi · 2023-07-31

**Summary Of Contributions:**

This paper investigates the use of a diagonal linear RNN model for modeling long sequences, as an alternative to both attention-based models as well as recently introduced continuous-time state space models. As these state space models such as S4 are continuous, which require later discretization for implementation, the suggestion here is to simplify matters by directly defining a discrete-time model. Fast training procedures based on FFTs are presented, analogous to methods used for S4. A number of benchmark tasks are introduced as well, which are used for benchmarking performance.

**Audience:**

Yes

**Claims And Evidence:**

No

**Requested Changes:**

* From the title and abstract of the paper, it seems the goal here is not just to introduce a new (simpler!) model, but also to aid in the *understanding* of the S4 model and other similar sorts of continuous state space models for sequence modeling. However, section 2 introduces this new model mathematically without ever explicitly relating it to the mathematical formulation of the continuous state space models it aims to help understand and simplify. Admittedly I am not very familiar with the related work, but I still think this makes it harder than necessary to understand the core contributions.

* It would be good to put these experimental results into a more clear context. Why compare against attention models at all if the main goal is to understand SSMs? Why compare on a novel set of benchmark tasks? Why not compare directly against SSMs?

**Strengths And Weaknesses:**

Strengths:

+ Clear motivation for the introduced model, with clear theoretical results
+ The presented model is written out in a clear and understandable manner, which could easily facilitate re-implementation

Weaknesses:

- The model here is presented as an RNN, and the relation to S4 is mentioned repeatedly but not really described mathematically. I'm very unfamiliar with this area but read the S4 paper as part of engaging with this review (I would have appreciated some description of it in related work / background). It's not clear to me exactly how this model differs from the discrete-time descriptions of S4, and in particular in terms of how the matrices are structured, it would be nice if this were made clear.
- If I understand correctly this model can be viewed as a simplification of the S4 model. Could this then be compared directly against S4 on the same long-sequence benchmark tasks in that paper? Why introduce new benchmark tasks for this model?

---

> ### Author Response · Authors · 2023-08-11
> **Response 1 to Reviewer jUMi**
>
> We thank the reviewer for the careful review and helpful suggestions.
>
> 1. ```Mathematical description of S4```: One of the main motivations for this work was in fact to free the reader from reading S4 in the first place, as it is **significantly more complex** and can be **cumbersome to read and implement**. Moreover, in **Section 2.1 (page 4) we include a detailed mathematical comparison of DLR to DSS which is a simpler variant of S4 and has been shown to be equally performant**. While in the DSS/S4D papers, the authors simplify S4 while maintaining its performance, they do *retain the continuous ODE and require discretization*. For this reason, we decided to use DSS as a representative baseline for continuous SSMs in all our experiments. DSS is same as S4D except it uses zero-order hold discretization instead of bilinear discretization and assumes the B parameter in the continuous state space ODE to be all 1’s vector.
>
> 2. ```Understanding SSMs```: Here the use of *“understanding”* is not intended to highlight the simplicity of DLR but **analyzing its capabilities** and pointing out the potential issues in convnets (including SSMs, DLR) from the point of view of in-context learning as their **weights are same for all inputs** unlike attention where the softmax-distributions are input-dependent. This is also why we evaluate on many tasks requiring in-context learning. E.g. on tasks such as **MIPS**, indeed the convnets struggle compared to attention (Table 1). On the other hand, on many tasks requiring in-context learning a deeper DLR stack is surprisingly performant. Here we are facilitating the **understanding of SSMs by testing them on various tasks requiring different long-range skills**.
>
> 3. ```Evaluation and additional experiments```: We agree with the reviewer that DLR can be compared directly against S4 on the tasks from the S4 paper. As per your suggestion, in **Table 4, page 10** we have **now included the evaluation of DLR vs S4 on Long Range Arena** (the main benchmark from S4 paper). We also evaluate on the 10-way raw speech classification task from S4 paper. On all these tasks **DLR reports competitive performance against the best performing variants of S4 and S4D**.
>
> |     |   LISTOPS |  TEXT   | RETRIEVAL |PATH-X | Speech Commands |
> |----|---|---|---|---|---|
> | S4|  59.6| 86.8 |90.0 |  96.4 |  98.1
> | DLR | 60.5| 86.7 | 89.1 | 94.5 | 97.1
>
> To clarify why we did not evaluate DLR on these tasks from S4 paper before is that we were specifically interested in sequence tagging tasks whereas Long Range Arena (main benchmark on which S4 was evaluated on) is sequence classification. I.e. in our tasks the model is provided a label per position whereas in LRA there is a label per sequence. Sparse supervision provides a weaker signal to the model but modern practices (language modeling, image modeling, score matching, etc) rely on pre-training with sequence tagging and NOT sequence classification and hence is a setting closer to the real-world cases. Moreover the lengths used in our proposed tasks are **significantly longer than used in LRA** allowing us to stress test the long-range expressivity of the model.

---

### Review · Reviewer_sEbY · 2023-08-06

**Summary Of Contributions:**

The authors redevelop the AR(1) method in the context of sequence learning and validate its effectiveness using toy scenarios.

**Audience:**

No

**Broader Impact Concerns:**

Not applicable.

**Claims And Evidence:**

No

**Requested Changes:**

I don't think the paper should be accepted at all.

**Strengths And Weaknesses:**

Strengths:
1. The authors develop the proposed method in details (with mathematical rigor).

Weakness:
1. The proposed method is well-known in statistics and signal processing --- although the authors invent a new name DLR for the method, but it is exactly the AR(1) model in statistics or IIR filter in signal processing.
2. For such a classic method, I had expected two directions that could contribute to ML community --- (1) develop its learning theory and (2) boost long-range capability for neural networks (e.g., Transformer) in practical setting (e.g., natural language processing). However, none of these exists in the current draft --- the method is used alone for toy problems.
3. Minor technical problem: when FFT is used to compute AR(1), there will be information leakage, i.e., the time series model is no longer casual --- prediction of the current step makes use of future information. The authors haven't discussed this implication.

---

### Review · Reviewer_TyAG · 2023-08-23

**Summary Of Contributions:**

This paper presents vanilla Diagonal Linear RNNs (DLR) that simplify sequence models based on linear state spaces (SSMs). The proposed model is fast to compute, due to the fact that only linear operations are involved, and shares some properties of linearity. Some experiments are conducted to show the performance.

**Audience:**

Yes

**Claims And Evidence:**

No

**Requested Changes:**

see my comments in the Weaknesses.

**Strengths And Weaknesses:**

**Strengths**

1. The paper is well-written and easy to follow and understand.

2. The proposed model is simple and easy to compute, and achieves some good performance on certain datasets.

**Weaknesses**

1. Generalization: One of my major concerns is whether the proposed method can be generalized to complex real-world datasets where the data structures are not so obvious. My understanding of the model is that it can be taken as a constrained polynomial mapping function (with no log and exp reparameterization, and i will discuss this part later), and it is well-known that such mapping functions will never work in some cases such as ring-like distributions (considering binary classification where positives inside form a disk and negatives are surrounding). As claimed in the abstract, the proposed models "struggle on tasks requiring many such kernels and especially when the desired sequence manipulation is context-dependent." This behavior can be interpretable based on my understanding, and I do not see how the proposed models can handle such cases (and the authors seem not to discuss this neither). This is a fundamental issue in the proposed models.

2. Reparameterization in Sec. 2.3: This part makes me confused about the contribution of the method. (1) If the model update was on $\lambda$ in the exp(log( )) formula, then all the properties on linearity of $\lambda$ discussed before would be lost; (2) If the model update was on exp(log( )) itself to preseve the linear properties, then the exploding/vanishing gradients in RNN training would still exist.

Another point is such reparameterization is quite similar to expRNN (Lezcano-Casado and Martinez-Rubio 2019), a parametrization stemming from Lie group theory through the exponential map. Unfortunately I did not see any discussion on the similarity and difference as well.

My understanding is that this reparameterization is the key to make the method working, but to me the contribution seems incremental to expRNN, without any discussion and comparison.

3. Experiments: I am not convinced by the experiments. I suggest the authors to compare different state-of-the-art RNNs such as expRNN on benchmark datasets to demonstrate the performance improvement. This is pretty standard in RNN papers, but missing in the current paper. Also, the code link does not work.

---

> ### Author Response · Authors · 2023-08-24
> **Response 1 to Reviewer TyAG**
>
> We thank the reviewer for the careful review and helpful suggestions.
>
> 1. ```Complex real-world datasets```: We clarify the misunderstanding. While a single DLR contextualization is linear in the inputs, the **DLR block is NOT linear** in the inputs and can model extremely complex data (below we show it can model natural language better than Transformers). Only the contextualization which occurs in the DLR part is linear. DLR layer is a generalization of traditional depthwise-convolutions and makes them global i.e. all output positions depend on all input positions unlike traditional convolutions where the kernels are highly local. As stated in **section 2.3 para 2** each **DLR block consists of GELU non-linearity after DLR layer** similar to traditional CNNs. Stacking multiple such blocks gives the model extremely good contextualization and the non-linearties allow it to model complex distributions.
>
> 2. ```struggle on tasks requiring many such kernels / context-dependent``` : This observation applies to single-layer DLR experiments which is a setup we used to understand the differences between a single DLR and single Transformer blocks. As shown in Table 1, a multi-layer DLR is indeed able to perform well on context-dependent tasks and outperforms Transformers on most tasks.
>
> 3. ```Reparameterization in Sec. 2.3```: The gradient update is on $\log_\lambda$. DLR in **not linear in lambda but in the INPUT u!!!!** i.e. $y_i = u_0*\lambda^i +... + u_i*\lambda^0$. We parameterize $\lambda$ as $\exp(-a^2 + i*b)$ where a, b are parameters on which gradient updates are applied. There are no exploding gradients as $|exp(-a^2)| < 1$ and no vanishing gradients if $|exp(-a^2)|$ is close to 1 i.e. a << 1. We hope this clears the misunderstanding regarding “linearity” of DLR.
>
> 4. ```Comparison with expRNN``` : As per your suggestion **we evaluate DLR on the p-MNIST task from expRNN paper** (in which MNIST images are flattened and then permuted using bit-reversal permutation). We find that DLR reports a significantly higher test accuracy over the best reported result in the expRNN paper
>
> |     |   pMNIST (test accuracy)|
> |----|---|
> | expRNN | 96.6
> | DLR | 98.8
>
> Moreover, RNNs such as **expRNN and SRU++ use non-linearities in the recurrence** are **NOT parallelizable across the length of the input**. The experiments in our work are on extremely long inputs on which these methods will be very slow due to the use of a for loop over time dimension. Secondly, these models will **suffer from vanishing gradients due to the use of non-linearities** across time dimension. The longest input length on which expRNN is evaluated in their paper is 2000 whereas we evaluate on significantly more complex tasks of length upto 500000.
>
> 5. ```Comparison on Long Range Arena``` : Based on suggestions from other reviewers we also **benchmarked DLR directly against the state of the art state space model S4** on the main Long Range Arena benchmark as well as 10-way raw human speech classification task. On all these tasks **DLR reports competitive performance against well-tuned best performing variants of S4 and S4D**.
>
> |     |   LISTOPS |  TEXT   | RETRIEVAL |PATH-X | Speech Commands |
> |----|---|---|---|---|---|
> | S4|  59.6| 86.8 |90.0 |  96.4 |  98.1
> | DLR | 60.5| 86.7 | 89.1 | 94.5 | 97.1
>
> 6. ```Comparison on Language Modeling``` : Based on suggestions from other reviewers we also performed large-scale **causal language modeling** on the Deepmind **PG-19 text corpus** and compare directly against Transformer with FLASH attention.  We tokenize individual books using T5 tokenizer and concatenate all tokens to get a sequence of 3 billion tokens. We then chunk these into sequences of size 4096 tokens. Each model has embedding dimension 128, model dimension 384 and 16 blocks where each feed-forward layer is a switch layer with 16 experts. For DLR, we simply replace the multi-head attention with DLR layer with N=512. We used a batch size of 128 with input length 4096 and trained for 60k training steps - well beyond the training budget suggested by Chinchilla scaling laws. For DLR learning rate was 1e-3 and for the Transformer baseline we tuned it over [1e-4, 3e-4, 1e-3]. **DLR not only reports a significantly lower perplexity compared to Transformer baseline** but was as fast as the highly optimized FLASH attention implementation while having a log-linear complexity and hence being more scalable to longer lengths as compared to attention.
>
> |     |   params | PG-19 test loss| steps/sec |
> |----|---|---|---|
> | Transformer| 318M | 3.71 |  1.05x
> | DLR | 312M | 2.88| 1x
>
> We hope this convinces the reviewer that a multi-layer DLR is a highly expressive & performant model capable of modeling complex real-world data.
>
> 7. ```Code link```: We are not allowed to provide a working code link to preserve anonymity. However, we assure the reviewer that DLR already has a public github repo which the reviewer can search online on their own accord.

---

> > ### Comment · Reviewer_TyAG · 2023-08-25
> >
> > I am confused about what exactly the implementation is for the proposed method. In the paper, the DLR is clearly defined in Eq. 1, with no nonlinear functions, but in your rebuttal "As stated in section 2.3 para 2 each DLR block consists of GELU non-linearity after DLR layer similar to traditional CNNs." If your statement in the rebuttal is true, then there should be one nonlinear function, at least, in the formula. See the definition of vanilla RNN.
> >
> > So could you clarify this first?

---

> > > ### Author Response · Authors · 2023-08-25
> > > **Response 2 to Reviewer TyAG**
> > >
> > > As **we state in Section 2.3 para 2**, a DLR block is computed as follows. Similar to S4, each DLR block receives a sequence $u$ of H-dimensional vectors and produces an output $y$ of the same shape. For this, first a DLR contextualization layer (Equation 4) is applied to (individual H dimensions of) $u$ to allow interaction between different positions across the time axis. Note that this is similar to depthwise-convolutions in CNNs except that DLR kernels are global whereas traditional CNN kernels are extremely local and restricted. This is followed by a residual connection from $u$ to $y$ and a GELU non-linearity followed by a position-wise linear projection $W$ to enable information exchange among the H coordinates. To summarize,
> > > ```
> > > y = W.GELU(u + DLR(u)) where W is a trainable parameter
> > > ```
> > > In the above equation, DLR is the contribution of our paper, whereas the formulation of the block is as used in the following previous works to be directly comparable to these works in our experiments
> > > 1. S4 [Gu et al, ICLR 2022]
> > > 2. DSS [Gupta et al NeurIPS 2022]
> > > 3. S4D [Gu et al NeurIPS 2022]

---

### Author Response · Authors · 2023-08-29
**Summary of updates made to the draft**

We thank the reviewers for their hard work and numerous helpful suggestions. Based on the reviews we have made the following changes to the draft:
1. ```Long Range Arena, Speech Commands```: We have **added a new sub-section 3.5** and **Table 4** in which we directly compare DLR with the best performing variants of S4 and S4D and report comparable performance to these SSMs on Long Range Arena and Speech Commands raw-speech classification - the main benchmark on which these SSMs were evaluated in previous papers.
|     |   LISTOPS |  TEXT   | RETRIEVAL |PATH-X | Speech Commands |
|----|---|---|---|---|---|
| S4|  59.6| 86.8 |90.0 |  96.4 |  98.1
| DLR | 60.5| 86.7 | 89.1 | 94.5 | 97.1

2. ```Causal Language Modeling on PG-19```: We have **added Table 5** in which we train 310M sized language models on PG-19 text corpus. **DLR reports comparable performance and throughput against Transformers having hardware-optimized attention implementation**. Interestingly, we find DLR to be **significantly more robust to the placement of layer norms** to which Transformers are highly sensitive to. Moreover, DLR enjoys $O(L)$ complexity during decoding time compared to $O(L^2)$ complexity in the case of Transformers.
|     |   params | throughput | train/test loss w/ post layer-norm | train/test loss w/ pre layer-norm|
|----|---|---|---|---|
| Transformer| 318M | 1x | DIVERGED | 2.52 / **2.63**
| DLR | 310M | 1x | 2.70 / **2.86** | 2.71 / 2.88

For Tables 4 and 5, we have also updated the Appendix with detailed descriptions of these experiments and their hyperparameters.

3. ```Non-linearities in DLR layer```: In Section 2.3, we now **more clearly state the use of non-linearities in the DLR layer** to avoid any confusion regarding the expressivity of a multi-layer DLR stack and its ability to model highly-complex data.

---

### Author Response · Authors · 2023-09-03
**Reminder to reviewers**

We thank the reviewers for their hard work and for helping us significantly improve the quality of our submission. Kindly have a look at the updated draft and let us know in case there's a need to perform further experiments, rewrite certain parts, etc.

---

### Decision · Action_Editors · 2023-10-19

**Recommendation:** Reject

**Comment:**

I appreciate the authors' efforts in the discussion period. However, multiple concerns remain regarding the novelty and technical contributions of this submission, leading all reviewers to recommend rejection:

- Insufficient technical description and direct comparison to closely related prior work to highlight novel contributions
- Concerns that method may resemble existing techniques without significantly advancing state-of-the-art

Comment from a reviewer:
>In particular questions about the expressivity of the model were not really addressed. I'm particularly concerned with the complete lack of response to reviewer sEbY. In their response to me, they pointed towards section 2.1 of the paper to address similarity to S4, but as far as I can tell the discretized model there is quite different from the eq (1) they claim it is similar to up to a scaling factor, as the expression for the x_t update in the DSS model involves a nonlinearity but in their discrete diagonal RNN model this is linear as well.


In paper, the authors claim that
>However, they invariably rely on discretization of a continuous state space, which complicates
 their presentation and understanding. In this work, we dispose of the discretization step,
 and propose a model based on vanilla Diagonal Linear RNNs (DLR).

In response, the authors state that
>Our contribution is not to re-invent AR(1) but to point out that one can directly start from diagonal linear RNNs, while retaining performance, rather than taking a more complicated approach of discretizing ODEs as done in the above listed works.

To summarize the reviewers' feedback and authors' responses, I suggest re-framing this work as a revisiting of AR(1) model for capturing long range dependencies and positioning the work as a study of AR(1) model for the problem, rather than proposing a new model. Given the similarities to AR(1) noted by reviewers, it is challenging to discern the contributions over existing models based on the evidence presented. I encourage the authors to clarify the technical advancements in a resubmission, as well as provide additional comparison to related models, in order to strengthen the claims in the work.

- Need for more thorough benchmarking and empirical analysis on standard tasks to evaluate capabilities
>Release the demo code: I do not think releasing any demo code without the authors' names will violate any policy for TMLR. This is very common in ML conferences where the authors can provide a demo code with a link so that the reviewers can access the code and verify some results. Personally, I do not buy the excuse from the authors, especially since I think Reviewer sEbY's comments are valid, and the authors need a way to convince us that their proposed method is good and significantly different from what sEbY commented. In the current format, I do not see the effort that the authors put to rebut such comments.

Given these significant reviewer concerns about novelty, technical depth, and adequacy of evaluation, I believe major revision is required before the work would be acceptable for publication.

**Audience:**

Yes

**Claims And Evidence:**

No

**Resubmission Of Major Revision:**

The authors may consider submitting a major revision at a later time.